# The Role of Selenium Nanoparticles in the Treatment of Liver Pathologies of Various Natures

**DOI:** 10.3390/ijms241310547

**Published:** 2023-06-23

**Authors:** Michael V. Goltyaev, Elena G. Varlamova

**Affiliations:** Institute of Cell Biophysics of the Russian Academy of Sciences, Federal Research Center “Pushchino Scientific Center for Biological Research of the Russian Academy of Sciences”, 142290 Pushchino, Russia; goltayev@mail.ru

**Keywords:** liver diseases, selenium, selenium nanoparticles

## Abstract

The liver is the body’s largest gland, and regulates a wide variety of physiological processes. The work of the liver can be disrupted in a variety of pathologies, the number of which is several hundred. It is extremely important to monitor the health of the liver and develop approaches to combat liver diseases. In recent decades, nanomedicine has become increasingly popular in the treatment of various liver pathologies, in which nanosized biomaterials, which are inorganic, polymeric, liposomal, albumin, and other nanoparticles, play an important role. Given the need to develop environmentally safe, inexpensive, simple, and high-performance biomedical agents for theragnostic purposes and showing few side effects, special attention is being paid to nanoparticles based on the important trace element selenium (Se). It is known that the metabolism of the microelement Se occurs in the liver, and its deficiency leads to the development of several serious diseases in this organ. In addition, the liver is the depot for most selenoproteins, which can reduce oxidative stress, inhibit tumor growth, and prevent other liver damage. This review is devoted to the description of the results of recent years, revealing the important role of selenium nanoparticles in the therapy and diagnosis of several liver pathologies, depending on the dose and physicochemical properties. The possibilities of selenium nanoparticles in the treatment of liver diseases, disclosed in the review, will not only reveal the advantages of their hepatoprotective properties but also significantly supplement the data on the role of the trace element selenium in the regulation of these diseases.

## 1. Introduction

The liver is an important organ that regulates a wide variety of physiological processes, including metabolism, protein, and lipid synthesis, production of enzymes that break down many toxic substances [1,2,3,4]. Therefore, it is extremely important to monitor the health of the liver and develop approaches for the treatment of this organ.

In recent decades, nanomedicine has become increasingly popular in the treatment of various liver pathologies. Nanomedicine is a branch of medicine in which nanosized biomaterials, including inorganic, polymer, liposomal, albumin, and other forms of nanoparticles have several significant advantages over traditional drug treatment options. These include targeted drug delivery, which ensures that drugs are delivered in a controlled manner and directly to the lesion site. In addition, the nanoparticles can form a barrier around the main substance, which prevents the inactivation of the therapeutic agent to its destination [1].

Inorganic nanoparticles usually consist of two layers: the core is represented by metal or metal oxide, and the outer surface consists of an organic layer that determines the optical, electrical, and magnetic properties of nanoparticles. Inorganic nanoparticles are widely used for the treatment of various liver pathologies. Therefore, it was shown that nanoparticles based on cerium oxide, gold, silver, and silica nanoparticles were used in the treatment of liver fibrosis [2,3,4,5,6].

Liposomal nanoparticles are microscopic spherical vesicles consisting of a core with an aqueous layer inside, and the core itself is enclosed within one or more phospholipid bilayers. Drug therapy based on liposomal nanoparticles, such as INF-α-loaded liposomes, vitamin-conjugated siRNAs, dexamethasone-loaded liposomes, etc. They show higher efficacy in the treatment of liver fibrosis than other nanoparticles used in clinical practice [7,8,9].

Polymer nanoparticles are colloidal in nature and are most often obtained by solvent evaporation, emulsification, and salting. They are used to deliver various biomolecules, such as sorafenib, nucleic acids, and nitric oxide. By polymer conjugation of sorafenib with poly (ethyleneglycol)-block-poly (lactic acid-co-glycolic acid) (PEG-PLGA), this antifibrotic agent was delivered to the systemic circulation, then to the liver, where it exerted an antifibrotic effect [10].

Albumin nanoparticles are biodegradable drugs often used in the treatment of liver pathologies, the main component of which is human or bovine serum albumin. Nanoparticles obtained by mixing berberine with bovine serum albumin were shown to inhibit the activation of liver stellate cells, preventing further fibrogenesis [11].

Nanomicelles, which are nanosized colloidal particles, consist of a hydrophobic core and a hydrophilic shell and are commonly used to deliver hydrophobic drugs to increase their concentration in systemic circulation [12,13,14].

Even though various variants of nanoparticles have been developed and tested in recent decades, such indicators as bioavailability, biocompatibility, biodegradability, toxicity, efficiency, and selectivity are very individual. This is primarily due to the nature of nanoparticles and their physicochemical properties. Due to the need to develop environmentally safe, inexpensive, simple, and high-performance biomedical agents used for theragnostic purposes and showing minor side effects, special attention is paid to nanoparticles based on the important trace element selenium (Se).

It is known that the metabolism of the trace element Se occurs in the liver, and its deficiency leads to the development of several serious diseases of this organ [5,6,7,8,9,10,11]. In addition, the liver is the depot of most selenoproteins, which can reduce oxidative stress, inhibit tumor growth and prevent other liver damage [12,13]. It has been shown that an increase in serum Se leads to an increase in the expression of selenoprotein P (SELENOP) and inhibition of pro-inflammatory factors and apoptosis [14]. It is known that SELENOP is secreted by the liver and transported to other organs, and its content in liver cancer cells is significantly reduced compared to healthy ones [15,16]. In addition, several drugs for the treatment of liver cancer are aimed at inhibiting the activity of selenium-containing oxidoreductases in tumor cells: thioredoxin reductases and glutathione peroxidases [17,18,19,20,21,22,23,24,25,26,27,28]. There are works demonstrating the role of other selenoproteins in the regulation of processes associated with pathological liver disorders [29,30,31]. However, there is still no complete picture of the understanding of the protective functions of Se and selenoproteins from various liver diseases, therefore, the development of new approaches to combat liver diseases is becoming more urgent.

Therefore, in recent decades, research using nanoparticles, in particular, selenium nanoparticles (SeNPs), has shown good prospects. With all the variety of liver pathologies (acute and chronic toxic damage, parasitic, infectious, oncological diseases), SeNPs in various studies show good results confirming their hepatoprotective and anticancer effects of both in vitro on liver cell culture and in vivo on animal models. Data were obtained on the direct cytotoxic effect of Se on liver cancer cells, on the use of SeNPs for the delivery of another anticancer agent to tumor cells, on the protection of liver cells both from hepatotoxins and to reduce the toxicity of classical anticancer drugs, as well as the protective effect of SeNPs when exposed to infectious and parasitic agents on the liver.

This review is devoted to the description of recent studies on the role of SeNPs in the treatment of liver pathologies of various natures, which will not only reveal the advantages of the protective properties of SeNPs of various natures but also significantly supplement the data on the role of the trace element selenium in the regulation of these diseases.

## 2. The Role of Nanomedicine in the Treatment of Liver Pathologies of Various Natures

### 2.1. The Role of SeNPs in the Treatment of Acute or Chronic Exposure to Hepatotoxic Agents

Hepatoprotectors are a pharmacotherapeutic group of heterogeneous drugs that stimulate the regeneration of hepatocytes and increase the resistance of the liver to pathological effects by increasing the activity of enzyme systems, therefore having a positive effect on the functioning of the liver.

To date, several works have demonstrated the hepatoprotective role of SeNPs. Thus, with an overdose of acetaminophen (para-acetylaminophenol, N-acetylpaminophenol, paracetamol, APAP), which is one of the most widespread over-the-counter antipyretic drugs with high toxicity, the protective properties of SeNPs in the liver of rats are described [32]. It is known that the danger of using APAP is associated with its hydroxylation by cytochrome P450 and the formation of the toxic metabolite N-acetyl-p-benzoquinonimine (NAPQI) [33]. The data obtained showed that the introduction of APAP was expected to lead to a significant increase in the level of liver enzymes in the blood of animals compared with the control group: alanine aminotransferase (ALT), aspartate aminotransferase (AST) and alkaline phosphatase (AP). However, the introduction of SeNPs significantly reduced the change in the level of these enzymes and contributed to a significant increase in the content of glutathione and glutathione peroxidase in the liver, a decrease in the concentration of malondialdehyde (MDA), catalase (CAT) and superoxide dismutase (SOD) activity, as well as a decrease in DNA fragmentation compared with the group of animals taking only APAP. Histological analysis showed the protective effect of SeNPs on hepatocytes: cells in the group taking APAP together with SeNPs had normal size and shape with normal morphology of the nucleolus and nuclear envelope, while in the group of rats taking only APAP, many cells contained pyknotic nuclei, various degenerative changes were observed in the cytoplasm of hepatocytes, such as hydropic dystrophy and fat infiltration. Transmission electron microscopy also showed the damaging effect of APAP on the nucleus and other cellular organelles of hepatocytes: chromatin condensation, thickening of the nuclear membrane, and the absence of clear boundaries of the nucleolus were observed. Similar disturbances in the morphology of cellular structures were restored when rats received SeNPs, which indicates that nanoparticles compensate for the negative effects caused by the action of APAP.

The protective effect of biogenic SeNPs in response to carbon tetrachloride-induced (CCl_4_) liver damage in mice has also been proven [34]. It is known that CCl_4_ causes oxidative stress and lipid peroxidation (POL) due to the generation of highly active radicals mediated by cytochrome P450 2E1 (CYP2E1), which ultimately leads to necrosis of liver cells [35]. The work used SeNPs obtained by reducing sodium selenite by the photosynthetic bacterium *Rhodopseudomonas palustris* (injected intragastrically, 200 mg/L of Se), and compared the effects of the obtained SeNPs (average size 80–200 nm) and sodium selenite. According to the results of the study, it was found that the pretreatment of SeNPs inhibited an increase in the activity of liver enzymes AST, ALT, AP, and lactate dehydrogenase (LDH) and simultaneously enhanced the activity of antioxidant enzymes SOD and CAT, reduced histopathological liver damage in mice treated with CCl_4_. Thus, it has been shown that SeNPs, by increasing the antioxidant capacity and inhibiting oxidative damage, can protect the liver from damage caused by CCl_4_.

To investigate the protective properties of SeNPs during autoimmune liver disease, a study was conducted [36] on a model of mouse liver damage induced by concanavalin A (ConA), a plant-derived protein that causes TNF-mediated apoptosis of hepatocytes [37]. SeNPs (average size 60 nm, zeta potential +31.2–+38.5 mV), obtained using ultrafiltration and stabilized with chitosan were used in the work [38]. The study was carried out on KM line mice, which were also injected intravenously with concanavalin A (20 mg/kg body weight) into the tail veins. SeNPs were injected daily through a gastric tube for 35 days. The obtained SeNPs demonstrated the ability to reduce the level of superoxide anion and hydroxyl radicals. The presence of SeNPs facilitated the course of hepatocyte necrosis induced by ConA and reduced the elevated levels of ALT, AST, and LDH in the blood serum of mice from the experimental groups. An increase in the activity of SOD, glutathione peroxidases (GPXs), and CAT was shown in mice treated with SeNPs.

In addition, the effect of SeNPs on the prevention of patulin-induced liver damage has been shown [39]. Patulin is a water-soluble and thermostable mycotoxin produced by mold fungi of the genera *Aspergillus*, *Penicillium,* and *Byssochlamys*, with toxicity to the gastrointestinal tract, kidneys, lungs, and liver [40,41]. SeNPs (about 133 nm in size, zeta potential +40.5 ± 2.34 mV), chemically obtained from sodium selenite using ascorbic acid and stabilized with chitosan, were used in the work. Both in vitro experiments were carried out on LO2 (human hepatocytes) and HEK 293 (human embryonic kidney cells) cells, and in vivo experiments on C57BL/6 mouse models were carried out. SeNPs (2 mg Se/kg body weight/day) were administered orally through a gastric tube for 5 days before patulin administration. The addition of 10 and 50 µM SeNPs12 h before the introduction of patulin significantly increased the percentage of viable cells (both LO2 and HEK 293) compared to cells treated with patulin alone, while preliminary exposure to sodium selenite showed no significant differences. Exposure to patulin caused a significant increase (approximately three times) in the intracellular level of ROS, while pretreatment of SeNPs reduced the level of ROS, weakening the patulin-mediated decrease in the activity of glutathione peroxidases. In in vivo experiments, the introduction of patulin significantly increased the level of ALT and AST in the liver, while the introduction of SeNPs returned the indicators to normal. The level of MDA increased up to three times compared to the control group and returned to normal with the preliminary introduction of SeNPs. After the introduction of patulin, the level of reduced glutathione (GSH) in liver tissue doubled compared to the control group, the level of glutathione disulfide (GSSG) in the liver tissue of the group receiving patulin was markedly increased, the redox status of glutathione (GSH/GSSG) decreased. Pretreatment of SeNPs weakened the patulin-induced violation of glutathione homeostasis but did not return the GSH concentration or the GSH/GSSG ratio to normal levels. The effect of patulin significantly reduced the activity of SOD and CAT in the liver. The introduction of SeNPs returned the SOD level to the control level but did not lead to a noticeable increase in CAT activity. Histological analysis showed that pretreatment of SeNPs before the introduction of patulin significantly reduced liver damage caused by patulin, although in some cells (mainly distributed near the central vein) there was slight vacuolization and lysis of the nucleus.

Several studies have demonstrated the therapeutic effect of SeNPs in liver poisoning with heavy metals. It was found that SeNPs obtained by the “green” method using lactic acid bacteria *Lactobacillus casei* had hepatotoxic effects caused by cadmium [42]. Both purified SeNPs and lacto-SeNP were used in the study. As a rule, treatment of the consequences of toxic exposure to heavy metals and, in particular, cadmium is based on chelation therapy using various chemical chelators, which can have several side effects, such as kidney overload, cardiac arrest, mineral deficiency, and anemia [43]. An alternative treatment in these cases may be the use of functional nutrition with SeNP content. SeNPs (with an average size of 80 nm, and a zeta potential of −22 mV) were used in the work, while experiments were carried out in vivo on CD1 mice, the animals first received cadmium in the form of CdCl_2_ orally. It is known that cadmium causes structural and functional disorders in cells due to an increase in lipid peroxidation, increases in the level of hepatic transaminases (ALT and AST), a significant decrease in the activity of catalase, glutathione peroxidase in the blood, and disrupts the permeability of cell membranes [44]. Treatment with both types of SeNPs caused a decrease in ALT and AST levels, as well as a dose-dependent increase in catalase activity compared to the group receiving only cadmium, while the activity of GPXs decreased both with the introduction of cadmium alone and cadmium with SeNPs. Histological analysis showed that exposure to cadmium caused degenerative changes in the liver: focal necrosis, hepatocyte lysis, and formation of pyknotic nuclei with condensed chromatin. The introduction of SeNPs practically prevented the appearance of pathological changes in the structure of the liver, with the best results obtained with the use of lacto-SeNPs. Immunohistochemical analysis of apoptosis markers: pro-apoptotic bax and anti-apoptotic bcl-2 revealed a noticeable increase in bax expression in the liver after exposure to cadmium and a dose-dependent decrease in the groups receiving SeNPs and lacto-SeNPs. On the contrary, bcl-2 expression was significantly reduced in the liver of mice in the cadmium-treated group and was dose-dependent restored in the SeNPs-treated groups, especially lacto-SeNPs. In addition, the expression level of *TNF-α*, *IL-6*, *NF-kB,* and *p65* genes was significantly increased in the cadmium-treated group (compared with the control), co-treatment of SeNPs and lacto-SeNPs led to a decrease in the expression of these genes.

In addition to cadmium, a study was conducted [45] on the therapeutic role of biogenic SeNPs or *Lacticaseibacillus rhamnosus* enriched with SeNPs (lacto-SeNPs) in the treatment of the consequences of toxic effects of lead on the liver. Lead is toxic to humans and animals, while among the organs in which lead accumulation is observed, the liver is in second place after the kidneys in terms of its concentration [46]. The classic way to quickly reduce the concentration of lead in the blood, as well as for cadmium, remains chelation therapy. The most commonly used chelating agents are ethylenediaminetetraacetic acid (EDTA), and dithiol preparations; however, they have side effects, since, in addition to lead, they bind trace elements, causing their deficiency in the body [47,48]. These effects lead to the need to search for alternative drugs, including nanoparticles. In this work, lactobacilli were used to restore Se from sodium selenite, while the synthesized nanoparticles had a size of 42.4 ± 10.5 nm and a zeta potential of −36.6 mV. Oral administration of lead acetate (150 mg/kg body weight per day) caused more than a 50-fold and 100-fold accumulation of lead in the blood and liver of mice of the C57BL/6 line, respectively, weight loss and liver enlargement were observed. Oral administration of lacto-SeNPs led to a decrease in the lead content in the blood and liver by 52% and 58%, respectively. The introduction of lacto-SeNPs restored significantly elevated levels of enzymes: ALT, AST, ALP, γ-glutamyltranspeptidase (GGT) to the control level, the use of SeNPs led to the restoration of only ALT and GGT levels. Elevated serum levels of low-density lipoproteins (LDL), high-density lipoproteins (HDL), total cholesterol, and triglycerides were also restored. The mechanism of the protective action of SeNPs when exposed to lead is the transfer of lead ions into an insoluble form and its subsequent excretion with feces. In addition, increased levels of glutathione in Se-enriched cells may also contribute to increased lead deposition.

The protective antioxidant role of SeNPs against the hepatotoxic action of acrylamide has been shown [49]. In the work, SeNPs with a size of 48–67 nm were used in vivo on a mouse liver model. The hepatotoxicity of acrylamide is due to a decrease in the level of GSH in the liver, and an increase in the serum level of liver enzymes (ALT, AST) and MDA [50]. The introduction of acrylamide caused a significant increase in the level of TNF-α with the joint administration of acrylamide and SeNPs, there was a significant decrease in the level of TNF-α. Exposure to acrylamide reduced the level of antioxidant enzymes: CAT, SOD, glutathione reductase (GR), glutathione-s-transferase (GST), while treatment with SeNPs caused an increase in the level of these enzymes. Histological examination showed the presence of fibrous connective tissue in the group receiving acrylamide, while there was an improvement in the histological picture with the combined administration of acrylamide and SeNPs compared with the group receiving only acrylamide.

Summary data on the hepatoprotective effect of SeNPs after exposure to hepatotoxic agents on the liver are presented in Table 1.

### 2.2. The Role of SeNPs in the Treatment of Infectious and Parasitic Liver Diseases

Several studies have shown the antiviral, antibacterial, and antiparasitic activity of SeNPs. The work [51] is focused on the study of the antimicrobial, antiviral, and mosquitocidal activity of biologically synthesized SeNPs (reduction of sodium selenite using a plant extract of *Portulaca oleracea*). Cytotoxicity against cancer cells (HepG2) and non-cancer cells (WI-38) was also investigated. The size of the obtained nanoparticles was 2–22 nm (average value 10.6 nm), and the zeta potential was −43.8 mV. The synthesized SeNPs showed a wide range of activity against Gram-positive bacteria, Gram-negative bacteria, and unicellular fungi: *Bacillus subtilis*, *Staphylococcus aureus*, *Pseudomonas aeruginosa*, *Escherichia coli*, *Candida albicans*, *Candida glabrata*, *Candida tropicalis, and Candida parapsilosis.*

The maximum inhibitory ability of nanoparticles was observed at a concentration of 300 µg/mL against these bacteria and fungi. The ability of SeNPs to protect Vera cells infected with *hepatitis A virus* (HAV) and *Coxsackie virus* (Cox-B4) was found, with a percentage protection of 70.25% and 62.5%, respectively, and the ability to increase cell viability to 84.2% and 76.5%, respectively. It is important to detect antiviral activity against HAV, since even with the presence of a vaccine, more than 114 million infections (symptomatic and asymptomatic) are recorded in the world per year [52]. It is possible that SeNPs suppressed the proliferation of viruses in *Vero* cells and inhibited the activation of apoptotic proteins by viruses inside host cells, as indicated in [53]. In addition, the cytotoxic effect of SeNPs on human hepatocellular carcinoma cells (HCC)—HepG2 and lung fibroblasts—WI-38 was analyzed. At a low concentration of SeNPs (31.25–62.5 µg/mL), cytotoxic effects in relation to normal cells were practically not observed, but a significant decrease in viability was characteristic of cancer cells. SeNPs had a strong cytotoxic effect in HepG2 cells, which was twice as high as their cytotoxicity in WI-38 cells. Additionally, the obtained SeNPs demonstrated larvicidal activity against *Culex pipiens* larvae. At high concentrations of SeNPs (50 mg/L), the mortality rate was 89.0 ± 3.1, 73 ± 1.2, 68 ± 1.4, and 59 ± 1.0% for larvae of I, II, III, and IV ages, respectively.

The effect of SeNPs on HepG2 cells infected with hepatitis B virus (HBV) has been shown [54,55]. Despite the appearance of a vaccine, viral hepatitis B still poses a danger to humans [56]. In addition, an adequate immune response does not develop in several patients after the introduction of a vaccine based on the hepatitis B virus surface antigen (HBsAg) [57]. The reason for this is a defect in the production of interferon-gamma (IFN-γ) and, consequently, a weakening of cellular immunity [58]. SeNPs with an average size of 45 nm and a zeta potential of −32.1 mV were used, they had a core-shell nanostructure with a width of about 30 nm and a length of about 200 nm. After the introduction of SeNPs into infected cells, the level of ALT and AST decreased, and the secretion of MDA decreased by 1.2 times. The anti-inflammatory effect of SeNPs in HBV-replicating HepG2 cells has been shown. After the introduction of SeNPs, the level of pro-inflammatory markers *TNF-α* and *TGF* decreased by 1.3 and 1.5 times, respectively, the level of *IL-8* decreased by 46% and *IL-2* by 43%, and SeNPs significantly protected cells from DNA damage.

The use of SeNPs as an adjuvant was also shown [59]: oral administration of nanoparticles together with subcutaneous immunization with an HBsAg-based vaccine formed a stable pattern of Th1 cytokines in a mouse model. Taking into account the previously shown property of SeNPs to increase the immune response of Th1 and trigger the production of cytokines, *IFN-γ*, *TNF-α*, *IL-12,* and *IL-2*, in one of the works it was proposed to use SeNPs as an adjuvant [60]. A significant increase in *IFN-γ* production was shown in the group receiving 100 and 200 µg of SeNPs and the vaccine, compared with the group receiving only the vaccine.

The hepatoprotective effect of the SeNPs-melatonin complex was found to protect against mouse liver damage caused by the Calmette-Guerin/lipopolysaccharide bacillus (BCG/LPS) [61]. According to literature data, lipopolysaccharides (LPS) are used as an experimental model of acute liver damage [62], while melatonin is proposed as a therapeutic agent for these liver lesions [63]. Injection of LPS causes acute liver damage with the release of ROS, NO, and pro-inflammatory cytokines [64]. Melatonin itself has antioxidant activity, but a synergistic effect is possible when used together with other antioxidants, including SeNPs. In the model group (BCG/LPS), severe liver damage was manifested with an increase in the level of plasma aminotransferases. When treated with the melatonin-selenium complex (5, 10, and 20 mg/kg), the levels of aminotransferases decreased significantly, and more strongly than in groups receiving only melatonin (10 mg/kg) or only Se (0.1 mg/kg). Histological analysis showed strong vacuolization of hepatocytes, necrotic changes in liver tissue, and stagnation in liver sinusoids with infiltration by inflammatory cells during BCG/LPS administration. Therapy with the melatonin-Se complex showed a decrease in the area and degree of necrosis, and infiltration of inflammatory cells, while no significant difference was found between the melatonin and selenium groups. When BCG/LPS was administered, the activity of GPXs and SOD was significantly inhibited, and the level of NO in plasma and liver tissue increased. Treatment with the melatonin-Se complex (10 and 20 mg/kg) increased the activity of GPXs and SOD and reduced the level of NO in plasma and liver, but Se (0.1 mg/kg) by itself did not affect NO production.

Several studies have shown scolicidal activity of SeNPs against echinococcal cysts [65,66]. Echinococcosis is caused by larval stages (metacestodes) of the tapeworm *Echinococcus granulosus* and leads to serious human health problems in several countries, as well as significant financial losses for agricultural producers. Infection with echinococcus occurs after ingestion of eggs by an intermediate host (human), then the eggs hatch in the upper parts of the small intestine, and an echinococcal cyst forms in the internal organs of the intermediate host: liver and lungs, and less often in muscles, kidneys, bones, spleen, and other organs [67]. Echinococcosis of the liver accounts for 50 to 70% of all cases of the disease. Surgical treatment of echinococcosis remains traditional; however, sometimes the cyst may be inoperable, in addition, secondary infection is possible with surgical intervention. Treatment with protoscolicidal cysts is an alternative method of treatment. Silver nitrate, cetrimide, hypertonic saline solution, and ethanol are used as such drugs, but they can cause toxemia and liver necrosis [68]. Nanoparticles can be used as new scolicidal materials with low side effects and high efficiency. When comparing the scolicidal effect of SeNPs and silver nanoparticles, a high scolicidal effect of SeNPs was established [65]. Biogenic SeNPs obtained by growing *Bacillus* sp. on a medium rich in Se^4+^ ions, a significant scolicidal effect was demonstrated in all studied concentrations (50–500 µg/mL) [66].

In addition, the preventive effect of SeNPs has been shown in acute toxoplasmosis [69] caused by *Toxoplasma gondii*, a universal intracellular parasite that affects a wide range of animals, in addition, it affects from 30 to 50% of the human population [70]. An effective combination for the treatment of toxoplasmosis is the simultaneous use of pyrimethamine and sulfadiazine; however, they cause certain side effects, such as osteoporosis, and sepsis, and have a teratogenic effect, especially in people with weakened immunity [71]. An alternative approach, in this case, may be the use of biogenic SeNPs. The experiments were carried out on NMRI mice, which were intraperitoneally injected with a virulent strain of *T. gondii* to create an animal model of acute toxoplasmosis. A significant difference in the survival rate of mice treated (5 and 10 mg/kg) compared with the group of animals not treated with SeNPs was shown. Additionally, the introduction of SeNPs significantly increased the expression of IFN-γ, TNF-α, IL-12, and NO-synthase, while the clinical biochemical parameters (ALT, AST, AP, bilirubin, creatinine) with the treatment with SeNPs (5 and 10 mg/kg) did not differ from the control values.

The anti-toxoplasmic effect of biogenic SeNPs obtained during the reduction of sodium selenite using *Streptomyces fulvissimus* has also been proven [72]. As a comparison drug, cotrimoxazole (a combination of sulfamethoxazole and trimethoprim 200/40 mg/5 mL) was administered orally to mice. Infected mice treated with SeNPs had a minimal number of parasites in liver and spleen swabs compared to the group that did not receive treatment. Scanning electron microscopy showed distortions in the morphology of tachyzoites (rough surface with small depressions and multiple protrusions, nanoscale deposits on the surface, and some tachyzoites deformed) in the group receiving SeNPs. The use of transmission electron microscopy showed that this group had the most extensive and pronounced morphological and structural changes in the form of destabilization of the membrane and leakage of its contents into the cytoplasm of the host cell. According to the results of the study, the therapeutic effect of SeNPs against *T. gondii* was not inferior to cotrimoxazole.

Summary data on the role of SeNPs in the treatment of infectious and parasitic diseases of the liver are presented in Table 2.

### 2.3. The Role of SeNPs in the Treatment of Liver Cancer

The work aimed at studying the role of SeNPs in the treatment of liver cancer can be divided into two groups: (1) finding ways to use SeNPs to reduce the negative effects of chemotherapy (chemoprotective activity), increasing the effectiveness of chemical anticancer drugs (chemosensitizing activity), helping SeNPs in the delivery of such drugs and (2) direct cytotoxic activity of SeNPs against cancer liver cells exceeding that of non-cancerous cells. In a study on the chemoprotective properties of SeNPs in hepatotoxicity in mice induced by cyclophosphamide, a widely used cytostatic antitumor drug of alkylating type of action, it was shown that intraperitoneal administration of cyclophosphamide caused a sharp increase in the level of ROS (by 53% in the liver), POL (by 48%), ALT (by 59%), AST (by 44%) [74]. Treatment also increased the level of GSH, GST, GPXs, SOD, CAT activity, and hemoglobin levels in the blood, and pre-administration of SeNPs significantly reduced ALT and AST levels. Improvements were also noticeable in histological sections: while hepatocellular necrosis was observed when exposed to cyclophosphamide, liver cells looked almost normal when pretreated with SeNPs.

The hepatoprotective effect of SeNPs against side effects caused by doxorubicin (DOX) has been described [75]. DOX, an anthracycline antibiotic is isolated from *Streptomyces peucetius* or *Streptomyces coeruleorubidus* and has antitumor activity. DOX intercalates DNA and inhibits macromolecular biosynthesis [76], while the main mechanism of DOX toxicity is the formation of ROS affecting macromolecules [73]. In addition, organ toxicity is also present, including in relation to the liver [77]. DOX was conjugated with SeNPs (42 nm), biosynthesized by reducing elemental Se from SeCl_4_ by fungus *Fusarium oxysporum*, the drugs were administered intraperitoneally to mice (DOX 5 mg/kg, SeNPs 0.5 and 1.5 mg/kg and conjugate 5 and 7 mg/kg). In the group of mice receiving conjugate (compared with the group receiving DOX), a decrease in DNA fragmentation, ROS levels, and the concentration of nitrates/nitrites, and an increase in the activity of antioxidant enzymes were observed.

In addition, the use of SeNPs as an agent for the delivery of ruthenium polypyridyl (RuPOP) has been described [78]. RuPOP has anticancer activity, including against liver cancer [79]. An important problem in cancer therapy is multidrug resistance, therefore, in addition to HepG2 cells, R-HepG2 cells with multidrug resistance were used in the study. Substances that ensure the delivery of the drug only to cancer cells, or deliver large concentrations of the active substance to cancer cells compared to normal ones, can help in solving this problem. Since the ability of SeNPs to act as carriers of anticancer drugs was previously shown [80,81], in this work, SeNPs conjugated with folic acid were used as a carrier of RuPOP (the method of preparation is chemical, the average size of SeNPs is 180 nm). Higher cytotoxicity and more selective cellular uptake of the SeNPs complex against R-HepG2 cells compared to HepG2 have been shown. Activation of apoptotic cell death of R-HepG2 and, to a lesser extent, HepG2 has been shown.

SeNPs are also used as a means for delivering small interfering RNA (miRNA, siRNA) to HepG2 cells [82]. The possibility of using gene therapy using siRNA technology in the treatment of cancer is being actively investigated [83], viruses are usually used as a means of delivery [84]; however, due to immunogenicity and the risk of insertion mutagenesis, methods of non-viral gene delivery have recently been developed [85]. SeNPs were obtained by reducing Se from sodium selenite with ascorbic acid, their conjugation was performed with hyaluronic acid, polyethylenimine (PEI), and then with miRNA. The size of the SeNPs was 70–180 nm, the zeta potential was about −22 mV, and after applying the positively charged PEI to the surface of the nanoparticles, the potential changed to +25 mV. The work used miRNA aimed at suppressing the *HES5* gene encoding the *HES5* transcription factor, which plays an important role in the initiation and development of cancer [86]. The resulting complex effectively suppressed the expression of *HES5* in vitro (inhibition of HepG2 cell proliferation and cell cycle arrest in the G0/G1 phase) and in vivo (inhibition of tumor growth by suppressing the expression of the *HES5* gene). In addition, there was no apparent toxicity to the main organs of mice.

Summary data on the role of chemoprotective and chemosensitizing activity of SeNPs are presented in Table 3.

The direct pro-apoptotic effect of SeNPs on HepG2 cells was also demonstrated in [87]. Both SeNPs (obtained by laser ablation, average size 100 nm, zeta potential −30 mV) and a complex of the obtained SeNPs with sorafenib were used. Sorafenib is a multikinase inhibitor that is effective even with progressive HCC [88], but has low bioavailability, rapid metabolism, and several side effects [89,90,91]. A pronounced pro-apoptotic effect on HepG2 cells of both SeNPs and SeNPs complex with sorafenib was shown, exceeding the effect of sorafenib. Sorafenib did not cause the generation of Ca^2+^ signals by HepG2 cells, whereas SeNPs and the complex (to a greater extent by 20–25%) led to the activation of the Ca^2+^ signaling system of cells.

Antitumor effect against HepG2 cells is also shown for SeNPs of biological origin obtained using CL90 polysaccharide extracted from lemon *Citrus limon* and stabilized by Tween-80 [92]. The average particle size was 79 nm. The antitumor activity of the obtained SeNPs was evaluated both in vitro (on HepG2 cells, causing their apoptosis in a dose-dependent manner) and in vivo (on a model of fish *Danio rerio*). It was shown that the CL90-SeNPscomplex, even in low concentrations, had a strong inhibitory effect on the proliferation, migration, and invasion of HepG2 cells.

Effective inhibition of HepG2 cells by the sesamol-polyethylene glycol—SeNPs complex has also been established [93]. Sesamol (3,4-methylenedioxyphenol) has anti-inflammatory [94] and anticancer [95] activity, but its use is limited by low stability and rapid elimination, therefore there is a need to develop complexes for delivery and increase its effectiveness. Polyethylene glycol (PEG) is an amphiphilic molecule and is widely used for drug delivery [96] and stabilization of nanoparticles. The diameter of the SeNPs-PEG complex ranged from 50 to 90 nm and slightly increased to about 100 nm with the addition of sesamol. The study showed a synergistic pro-apoptotic effect of the sesamol-PEG-SeNPs complex on HepG2 cells, the apoptotic activity of the complex was associated with the suppression of BCL-2, poly(ADP-ribose)-polymerase (PARP), increased activity of BAX and release of cytochrome c into the cytosol. Ultrasmall nanoparticles were obtained by the interaction of gray Se with PEG, with an average diameter of about 5 nm and a hydrodynamic diameter of about 28.7 nm [97]. PEG-SeNPs showed dose-dependent cytotoxicity against both HepG2 and R-HepG2 cells, while the particles showed much lower cytotoxicity relative to normal cells (human kidney HK-2 cells). Induction of apoptosis in R-HepG2 and HepG2 cells was recorded, which was confirmed by DNA fragmentation and nuclear condensation, as well as mitochondrial dysfunction.

SeNPs (5, 10, and 20 µg/mL) obtained biologically during the reduction of Se from sodium selenite with an aqueous extract of hawthorn *Crataegus hyphens* were also contributed to the apoptosis of HepG2 cells [98]. The average size of the obtained SeNPs was 113 nm, and the zeta potential was-24.5 mV. Treatment with SeNPs increased intracellular ROS levels, decreased mitochondrial membrane potential, and induced an increase in the level of caspase-9 and a decrease in Bcl-2.

In addition, cytotoxicity against HepG2 is also shown for the SeNPs complex with alcohol-soluble astragalus polysaccharide (AASP) isolated from the stems or dried roots of *Astragalus membranaceus*, and having biologically active effects [99]. It is known that AASP can inhibit the growth of solid H22 tumors in mice by increasing the level of serum cytokines (TNF-α, IL-2, and IFN-γ) and the activity of immune cells (macrophages, lymphocytes, and NK cells), therefore inducing apoptosis of tumor cells [100]. SeNPs were obtained by reducing sodium selenite with ascorbic acid and had an average size of about 50 nm and a zeta potential of −37.08 mV. This nanocomplex activated mitochondrial-mediated apoptosis in HepG2 cells and induced the accumulation of intracellular ROS, increasing the Bax/Bcl-2 ratio and promoting the release of cytochrome C.

The authors of the study [101] conducted a comparative analysis of the effects of chemically synthesized SeNPs of different sizes (35 and 91 nm) in vitro (Tca8113 cell line, squamous cell carcinoma of the human tongue) and in vivo (H22 cell line, mouse HCC). SeNPs of large and small sizes at a dosage of 0.7 mg Se/kg inhibited the growth of H22 cancer cells by 82% and 99%, respectively. Thus, small-sized SeNPs showed significantly greater efficiency, obviously due to the larger surface area. It is shown that the recovery of SeNPs depends on the concentration of GSH and leads to the formation of ROS.

When studying the properties of a Se-substituted hydroxyapatite nanocomplex with a size of 183 ± 6 nm, antitumor activity against HCC was established, which was shown by the example of a human HCCLM9 cell line with a high metastatic potential [102]. Hydroxyapatite is known to be the main inorganic mineral in human and animal hard tissues, including bones and tooth enamel, and it was previously found that its nanoparticles induce apoptosis and inhibit HCC cells [103]. Nanocomplex treatment significantly improved the survival of naked mice injected with HCCLM9 cells, and increased ALT, AST, and LDH levels decreased. In the group of mice receiving Se-substituted hydroxyapatite, metastases were not observed, which were visible in the control group and the group receiving only hydroxyapatite.

The therapeutic effect of SeNPs on rat HCC caused by N-nitrosodiethylamine (NDEA) was described in [104]. It is known that N-nitrosodiethylamine has mutagenic and carcinogenic activity, including against the liver [105]. HCC in rats was induced by oral administration of NDEA, then treated with DOX, SeNPs, and their combination. SeNPs were obtained by the chemical method: reduction of sodium selenite with GSH in the presence of bovine serum albumin, the size of the obtained particles was 20–60 nm. The rate of necrosis/apoptosis in tumor cells was maximal in the group receiving SeNPs, the expression of Akr1b10 and ING3 genes significantly increased compared to the group not receiving treatment. On the other hand, the expression of the Foxp1 gene was significantly reduced in the treatment of SeNPs. The data obtained are also confirmed by histopathological examination.

The suppression of HCC by Se and quercetin nanoparticles caused by thioacetamide (TAA), which is an organosulfur compound (C_2_H_5_NS), capable of causing acute or chronic fibrosis and cirrhosis of the liver in experimental animals, was described [106,107]. Administration of TAA to rats causes hepatic encephalopathy, metabolic acidosis, increased transaminase levels, abnormal coagulopathy, and centrilobular necrosis. It is known that TAA undergoes bioactivation in the liver by cytochrome P450 (CYP450 2E1), which leads to the formation of TAA-S-oxide and TAA-S-dioxide. TAA-S-dioxide induces oxidative stress through POL in the hepatocellular membrane [108]. The processes described above can lead to the development of HCC [109,110].

In addition, it was shown to overcome resistance to sorafenib in rats with HCC induced by TAA using 60 nm SeNPs, which were chemically obtained from sodium selenite using ascorbic acid and chitosan [106]. In the group receiving TAA, the levels of AST, ALT, GGT, and MDA increased and the level of GPXs was reduced. Oral administration of SeNPs, sorafenib, and their combination largely eliminated the negative effects caused by TAA. Treatment with a complex of SeNPs + sorafenib returned the level of all the above enzymes to normal, this group was superior to the group receiving only sorafenib in terms of ALT and AST. The work also showed the ability of SeNPs to stop the inhibition of p53 expression induced by HCC, activate the apoptotic pathway of CASP3 and Bax and inhibit the anti-apoptotic effect of Bcl2. Combination therapy with SeNPs + sorafenib reduced the resistance of HCC cells to sorafenib by activating the mTOR and NF-kB pathways, as well as reducing CD34 levels. This therapy has reduced angiogenesis and tumor metastasis.

Treatment with SeNPs, obtained by hydrothermal method from sodium selenite and quercetin, which has antioxidant and anti-inflammatory properties [107], showed a decrease in ALT, AST, and total bilirubin levels, increased after induction of HCC. Additionally, the introduction of SeNPs prevented an increase in IL-33, IL-1ß, and IL-6, lowered the level of MDA in the liver, and increased the level of GSH and GPXs compared to other groups. Treatment with both SeNPs and SeNPs + quercetin suppressed the progression of HCC in rats due to increased oxidative stress and prevention of dysregulation of the p53/β-catenin/cyclin D pathway.

Summary data on the role of SeNPs in the treatment of liver cancer are presented in Table 4.

Summary data on the role of SeNPs in the treatment of liver pathologies of various natures are presented in Figure 1.

## 3. Discussion

Nanomedicine for the treatment of liver pathologies has several significant advantages over traditional medicine, providing targeted drug delivery to tumors due to the effect of increased permeability and retention, including the delivery of more than one therapeutic agent, which is important for combination therapy. In addition, different variations of the target ligand on the surface of nanoparticles can contribute to a prolonged effect, which increases the effectiveness and reduces the side effects of drugs [16,17,18,19,20]. Due to the need to develop environmentally safe, inexpensive, simple, and high-performance biomedical agents used for theragnostic purposes and showing minor side effects, special attention is paid to nanoparticles based on the important trace element selenium (Se).

It is known that the metabolism of the trace element Se occurs in the liver, and its deficiency leads to the development of several serious diseases of this organ [5,6,7,8,9,10,11]. In addition, the liver is the depot of most selenoproteins, which can reduce oxidative stress, inhibit tumor growth and prevent other liver damage [12,13].

This review describes in sufficient detail the hepatoprotective role of SeNPs from hepatotoxic agents such as acetaminophen [32], carbon tetrachloride [34], concavalin A [36], patulin [39], heavy metals, for example, cadmium [42], lead [45]. The hepatoprotective mechanism of action of SeNPs is based mainly on their antioxidant potential. With nanoparticle therapy, there is a decrease in the level of liver enzymes in the blood (primarily ALT, AST), increased as a result of the action of hepatotoxin. In some studies, the ALT/AST level is shown to return to normal, in some it does not reach the control values, but everywhere it is significantly lower than the level of enzymes in the group that received only the toxin. In addition, SeNP therapy reduces the level of AP, MDA, and LDH in the blood, and increases the activity of antioxidant enzymes: SOD, GPXs, and CAT. Studies on liver cells have shown the ability of SeNPs to reduce the level of DNA fragmentation in hepatocytes, the level of intracellular ROS decreases, and the ratio of GSH/GSSG returns to normal. Treatment with SeNPs resulted in a decrease in the expression of *TNF-α*, *IL-6*, *NF-kB,* and *BAX* genes, an increase in the expression of *BCL-2*, and its return to normal.

Several studies have shown the antiviral, antibacterial, and antiparasitic activity of SeNPs in the liver [51,54,55,61,65,66,69,72]. The synthesized SeNPs showed a wide range of activity against Gram-positive and Gram-negative bacteria, unicellular fungi. SeNPs suppressed the proliferation of viruses and inhibited the activation of apoptotic proteins, had an anti-inflammatory effect, reducing the level of TNF-α, TGF, IL-8, and IL-2, and protected cells from DNA damage. Sometimes SeNPs are used as an adjuvant [59]: oral administration of SeNPs together with subcutaneous immunization with an HBsAg-based vaccine formed a stable pattern of Th1 cytokines in a mouse model. Several studies have shown scolicidal activity of SeNPs against echinococcal cysts [65,66].

Many works are devoted to the study of the role of SeNPs in the treatment of oncological liver diseases [74,75,78,82,87,92,93,97,98,101,102,104,106,107]. SeNPs in cancer cells, as a rule, had a pro-oxidant effect, increasing the production of ROS, pro-inflammatory effect, increasing the expression of serum cytokines (TNF-α, IL-2, and IFN-γ) and the activity of immune cells (macrophages, lymphocytes, and NK cells), caused apoptosis, inducing an increase in the level of caspase, the ratio of PAH/Bcl-2, contributing to the release of cytochrome C, etc.

This review shows that the effectiveness and mechanism of action of SeNPs, first, depends on such important physicochemical parameters as size, zeta potential, and method of production. Due to their unique surface activity and dispersion, SeNPs have higher biological activity, high catalytic efficiency and bioavailability, and lower toxicity. The possibility of manufacturing nanoparticles free of any toxic or dangerous substances is a very difficult task, especially for applications in nanomedicine. In addition, one of the important unresolved problems in the field of biomedicine in liver disease, especially in HCC, is the rapid removal of nanostructures from the body. Hepatocytes in the liver endocyte of nanoparticles can release them either into the bloodstream or into bile. In addition, Kupfer cells, which make up 80–90% of all macrophages of the body, are responsible for most of the phagocytic activity of the liver. These cells, together with monocytes and macrophages of the spleen, circulating in the blood, make up a system of mononuclear phagocytes, which is responsible for the sequestration of more than 95% of nanoparticles, which eventually do not reach their goal [21,22]. Therefore, new developments based on nanoparticles for the treatment of various liver pathologies are very relevant, and take into account a certain dual role of the liver, as an organ performing purification from nanostructures, and a target organ for targeted drug delivery [15]. SeNPs are valuable nanoplatforms with many desirable characteristics for clinical use. Due to the possibility of accurate calibration and rational modification of their physicochemical properties, the new SeNPs are particularly attractive as therapeutic agents that are easily tolerated in the body and provide stability in the physiological microenvironment of target tissues. Even though diagnostics and therapy based on SeNPs are in the early stages and are preparing to move to clinical trials, the effectiveness of the therapeutic hepatoprotective action of SeNPs has been repeatedly proven through various independent studies, which opens up high prospects for the use of drugs based on them in medical practice.

## 4. Conclusions

The results of this review study show the high potential of SeNPs in the treatment of liver diseases caused by various agents. Many studies have shown the hepatoprotective, antiviral, and antiparasitic activity of SeNPs. We have demonstrated the ability of selenium nanoparticles to protect the liver from classical anticancer drugs and facilitate their delivery, as well as directly affecting hepatocellular carcinoma cells, causing their apoptosis. Like other forms of selenium, SeNPs have both antioxidant and prooxidative activity: on the one hand, the presence of selenium increases the expression of selenium-containing antioxidant enzymes, which underlies its hepatoprotective properties, on the other hand, selenium metabolism itself causes the formation of ROS, and in cancer cells to a greater extent than in normal ones, while, unlike other forms of selenium, SeNPs have less toxicity. This feature, as well as the ability of SeNPs to induce apoptosis in liver cancer cells, makes SeNPs a promising therapeutic drug both in pure form and, especially, in combination with other anticancer agents.

## Figures and Tables

**Figure 1 ijms-24-10547-f001:**
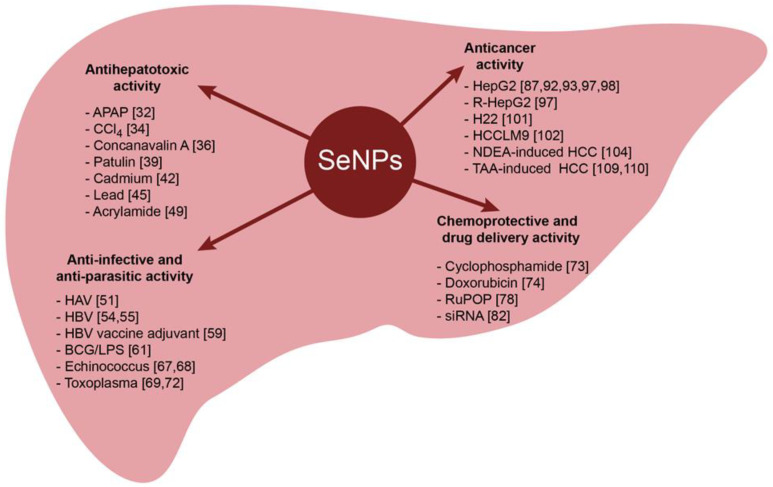
The role of SeNPs in the treatment of liver pathologies of various natures.

**Table 1 ijms-24-10547-t001:** Summary data on the hepatoprotective effect of SeNPs after exposure to hepatotoxic agents on the liver.

Hepato-Toxic Agents	Hepatotoxic Effects of Various Agents	Hepatoprotective Effects of SeNPs	Ref.
APAP	ALT, AST, AP, MDA GPXs, GSH, CAT, SOD - irregular shape of cells - thickening of the cell membrane–pyknotic nuclei - lack of clear boundaries - DNA fragmentation - chromatin condensation	ALT, AST, AP, MDA CAT, SOD, GPXs - cells of the correct shape and size - cell membrane of normal thickness - correct morphology of nuclei and nucleoli - absence of DNA fragmentation	[32]
CCL_4_	ALT, AST, AP, LDH, ROS CAT, SOD POL, necrosis	ALT, AST, AP, LDH CAT, SOD necrosis	[34]
Con A	ALT, AST, LDH, ROS CAT, SOD, GPXs TNF-mediated apoptosis and necrosis	ALT, AST, LDH, superoxide anion, hydroxyl radicals CAT, SOD, GPXs TNF-mediated apoptosis and facilitate necrosis	[36]
Patulin	ALT, AST, MDA, GSH/GSSG GSH, GSSG, GPXs, SOD, CAT vacuolization and lysis of nuclei cell viability	ALT, AST, MDA, ROS SOD, CAT, GPXs, GSH/GSSG vacuolization and lysis of nuclei cell viability	[39]
Cadmium	ALT, AST CAT, GPXs POL BAX, TNF-α, IL-6, NF-kB, p65 BCL-2 - causes structural and functional disorders in cells - disrupts the permeability of cell membranes - focal necrosis - hepatocyte lysis - formation of pyknotic nuclei with condensed chromatin	ALT, AST CAT POL BAX, TNF-α, IL-6, NF-kB, p65 BCL-2 - restore violations of the structure and functions of cell	[42]
Lead	ALT, AST, AP, GGT LDL, HDL	ALT, GGT, HDL, LDL, total cholesterol triglycerides GSH	[45]
Acrylamide	ALT, AST, MDA, TNF-α GSH, SOD, CAT, GR, GST fibrous connective tissue	ALT, AST, MDA, TNF-α GSH, SOD, CAT, GR, GST fibrogenesis	[49]

Abbreviations: APAP—acetaminophen; ALT—alanine aminotransferase; AST—aspartate aminotransferase; AP—alkaline phosphatase; MDA—malondialdehyde; GPXs—glutathione peroxidases; GSH—glutathione; CAT—catalase; SOD—superoxide dismutase; CCL_4_—tetrachloride-induced; LDH—lactate dehydrogenase; ROS—reactive oxygen species; POL—lipid peroxidation; Con A—concanavalin A; TNF—tumor necrosis factor; GSSG—glutathione disulfide; BAX—BCL2 associated X, apoptosis regulator; IL—interleukin; NF-kB—nuclear factor kappa B; p65—oncofetal protein; BCL-2—B-cell leukemia/lymphoma-2-alpha protein; GGT-γ—glutamyl transpeptidase; LDL—low-density lipoproteins; HDL—high-density lipoproteins; GR—glutathione reductase; GST—glutathione-s-transferase. Indicators that increased are highlighted in red, indicators that decreased are highlighted in blue, and indicators that were normalized are highlighted in green.

**Table 2 ijms-24-10547-t002:** Summary data on the role of SeNP in the treatment of infectious and parasitic diseases of the liver.

Infection/ Parasitic Agents	Hepatotoxic Effects of Various Agents	Hepatoprotective Effects of SeNPs	Ref.
HAV	-	- antiviral activity (anti-HAV, anti-Cox-B4) - larvicidal activity (*Culex pipiens*)	[51]
HBV	ALT, AST, MDA TNF- α , TGF, IL-8, IL-2	ALT, AST, MDA TNF- α , TGF, IL-8, IL-2 - antiviral activity (anti-HBV)	[54,55]
HBV vaccine adjuvant	-	IFN-γ	[59]
BCG/LPS	ROS, NO, ALT, AST GPXs, SOD	ALT, AST, NO GPXs, SOD	[61]
Echinococcus	-	- scolicidal activity	[66,73]
Toxoplasma	-	IFN-γ, TNF-α, IL-12, NO-synthase ALT, AST, AP, bilirubin, creatinine	[69,72]

Abbreviations: HAV—hepatovirus A; HBV—hepatovirus B; BCG/LPS—Calmette-Guerin/lipopolysaccharide bacillus; Cox-B4—coxsackie virus; ALT—alanine aminotransferase; AST—aspartate aminotransferase; AP—alkaline phosphatase; MDA—malondialdehyde; GPXs—glutathione peroxidases; SOD—superoxide dismutase; ROS—reactive oxygen species; TNF—tumor necrosis factor; TGF—transforming growth factor; IL—interleukin; IFN—interferon. Indicators that increased are highlighted in red, indicators that decreased are highlighted in blue, and indicators that were normalized are highlighted in green.

**Table 3 ijms-24-10547-t003:** Summary data on the role of chemoprotective and chemosensitizing activity of SeNPs.

Chemotherapy/ Drug Delivery Agent	Hepatotoxic Effects of Various Agents	Hepatoprotective Effects of SeNPs	Ref.
Cyclophosphamide	ALT, AST, MDA, ROS, POL GPXs, GST, SOD, CAT - hepatocellular necrosis - DNA damage and chromosomal aberration	ALT, AST GSH, GST, GPXs, SOD, CAT - hemoglobin level	[74]
DOX	ROS, MDA DNA fragmentation BCL-2	ROS, MDA DNA fragmentation concentration of nitrates/nitrites BCL-2	[75]
RuPOP	- apoptosis induction	ROS - apoptosis induction (activating p53 and MAPKs pathways)	[78]
siRNA	-	- non-viral gene delivery	[82]

Abbreviations: DOX—doxorubicin; RuPOP—ruthenium polypyridyl; siRNA—small interfering RNA; ALT—alanine aminotransferase; AST—aspartate aminotransferase; MDA—malondialdehyde; GPXs—glutathione peroxidases; GSH—glutathione; CAT—catalase; SOD—superoxide dismutase; ROS—reactive oxygen species; POL—lipid peroxidation; BCL-2—B-cell leukemia/lymphoma-2-alpha protein; GST—glutathione-s-transferase. Indicators that increased are highlighted in red, indicators that decreased are highlighted in blue, indicators that were normalized are highlighted in green.

**Table 4 ijms-24-10547-t004:** Summary data on the role of SeNPs in the treatment of liver cancer.

Liver Cancer Cell Line/Carcinogenic Agent	Hepatotoxic Effects of Various Agents	Hepatoprotective Effects of SeNPs	Ref.
HepG2	-	ROS, BAX, caspase-9 BCL-2 - HCC cells apoptosis - HCC cells cytotoxicity - DNA fragmentation, nuclear condensation - mitochondrial dysfunction	[87,92,93,97,98]
R-HepG2	-	- HCC cells apoptosis - HCC cells cytotoxicity - DNA fragmentation, nuclear condensation - mitochondrial dysfunction	[97]
H22	-	ROS, TNF-α, IL-2, IFN-γ, BAX BCL-2 - HCC cells apoptosis	[101]
HCCLM9	-	ALT, AST, LDH - HCC cells cytotoxicity	[102]
NDEA	ROS, MDA GR, GPXs, SOD, GSH	Akr1b10, ING3 Foxp1 - necrosis/apoptosis in HCC cells	[104]
TAA	POL, AST, ALT, GGT, MDA IL-33, IL-1ß, IL-6 GPXs	AST, ALT, GGT, MDA, GPXs IL-33, IL-1ß, IL-6 Bax, NF-kB Bcl2	[106,107]

Abbreviations: NDEA—N-nitrosodiethylamine induced hepatocellular carcinoma; TAA—thioacetamide-induced hepatocellular carcinoma; Akr1b10—aldo-keto reductase 1B10; ING3—inhibitor of growth family member 3; Foxp1—forkhead box P1; ALT—alanine aminotransferase; AST—aspartate aminotransferase; MDA—malondialdehyde; GPXs—glutathione peroxidases; GSH—glutathione; SOD—superoxide dismutase; LDH—lactate dehydrogenase; ROS—reactive oxygen species; POL—lipid peroxidation; TNF—tumor necrosis factor; BAX—BCL2 associated X, apoptosis regulator; IL—interleukin; NF-kB—nuclear factor kappa B; BCL-2—B-cell leukemia/lymphoma-2-alpha protein; GGT-γ—glutamyl transpeptidase; GR—glutathione reductase. Indicators that increased are highlighted in red, indicators that decreased are highlighted in blue, indicators that were normalized are highlighted in green.

## Data Availability

The data presented in this study are available on request from the corresponding author.

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
