# Peer review of "The Role of Selenium Nanoparticles in the Treatment of Liver Pathologies of Various Natures"

_ijms, 2023, doi:10.3390/ijms241310547_

Round 1

Reviewer 1 Report

The manuscript entitled “The role of selenium nanoparticles in the treatment of liver pathologies of various nature” by M.V. Goltyaev and E.G. Varlamova is devoted to the analysis of literature data regarding the molecular effects of selenium nanoparticles on liver fibrosis and cancer. The review may arouse great interest among readers. The manuscript may be accepted for publication after some improvements.

Comments

1) It is necessary to strictly observe the International Gene and Protein Nomenclature Guidelines for writing the names of genes and proteins. Gene names should be in italics.

2) Tables very often contain only one reference to literary data. It is necessary to additionally indicate a few more studies and/or make sure that the submitted works are unique in their kind.

3) Figure 1 as presented misleads the reader. The lobes of the liver and the schematic image of the liver as a whole are not informative. It seems that the effects presented in the form of arrows are observed only in certain regions of the liver. The figure should be simplified and presented in a more informative way.

The quality of the English language is good, but the text needs additional proofreading for typos.

Author Response

Dear Reviewer!

The authors of the manuscript are grateful to you for your work and valuable comments, which we tried to correct.

  1. All gene names in the text are in italics.
  2. In the table, we sometimes indicate the only sources of literature, since at present there is very little information on the role of selenium nanoparticles in various liver diseases. We once again checked whether there were new publications on this topic, but found nothing. Therefore, if you do not mind, we will leave in some places in the table a link to a single source.
  3. Fig.1. simplified, removed the image of the liver lobes

Reviewer 2 Report

This is a fascinating review that covers the therapeutic abilities of selenium nanoparticles to treat various liver pathologies. This is highly relevant and an up-and-coming area in the selenium field. For the most part, the review is well-written and comprehensive. I only have some minor comments for the authors to consider and some slight grammatical corrections. 

Specific comments:

1.     Lines 29-31 – could change to “metabolism, protein, and lipid synthesis, and production of enzymes that break down many toxic substances”. No need to include etc. 

2.     Lines 34-36 – can change to “Nanomedicine, a branch of medicine in which nanosized biomaterials including inorganic, polymer, liposomal, albumin, and other forms of nanoparticles have a number of significant advantages over traditional drug treatment options.” 

3.     The introduction could be condensed. It is not necessary to go through all the different forms of nanoparticles that could be covered in a mini-section elsewhere in the review as it unnecessarily lengthens the introduction and causes the main points to get somewhat lost. 

4.     Lines 115-118 – Run-on sentence. Consider revising. 

5.     Lines 219-220 – Revise to “liver, with the best results obtained with the use of lacto-SeNPs”.

6.     Lines 235-236: “The most commonly used chelating agents are ethylenediaminetetraacetic acid (EDTA), and dithiol preparations, however…”

7.     The figure is very nice and summarizes the review very well. 

8.     Line 628 – Change ; to , 

9.     Other minor grammatical errors throughout the manuscript

The quality of the English language is for the most part, excellent. I cited some of the specific grammatical errors that I noticed. Still, I did not mention all of them so the authors may want to go over the manuscript thoroughly and edit the minor grammatical errors.  

Author Response

Dear Reviewer!

The authors of the manuscript are grateful to you for your work and valuable comments, which we tried to correct.

  1. In lines 29-31 at the end of the sentence was removed, etc.
  2. In lines 34-36, the sentence was replaced with the option you proposed.
  3. If you do not categorically object, we would like to leave in the introduction the information that concerns a brief description of all nanocompounds used for the treatment of liver diseases. In our opinion, this is necessary in order to first make it clear to the reader what other nanostructures are used in the world to treat liver diseases. Next, we bring the reader to an understanding of why selenium nanoparticles are better suited for these purposes and point out their advantages. However, we have significantly reduced the description of these nanostructures.
  4. In lines 115-118, a correction was made.
  5. In lines 219-220, a correction was made.
  6. In lines 235-236, a correction was made.
  7. In line 628, a correction was made.
  8. We checked the entire text for errors and typos, and made the appropriate changes.
